# Four New Species of *Tomentella* (Thelephorales, Basidiomycota) from Subtropical Forests in Southwestern China

**DOI:** 10.3390/jof10070440

**Published:** 2024-06-21

**Authors:** Ya-Quan Zhu, Xue-Long Li, Dong-Xue Zhao, Yu-Lian Wei, Hai-Sheng Yuan

**Affiliations:** 1CAS Key Laboratory of Forest Ecology and Management, Institute of Applied Ecology, Chinese Academy of Sciences, Shenyang 110164, China; 14740578353@163.com (Y.-Q.Z.); zdx19970201@126.com (D.-X.Z.); weiyulian@iae.ac.cn (Y.-L.W.); 2University of Chinese Academy of Sciences, Beijing 100049, China; 3Institute of Edible Fungi, Liaoning Academy of Agricultural Sciences, Shenyang 110161, China; lixuelong2008@126.com

**Keywords:** ectomycorrhizal fungi, phylogeny, subtropical mixed forest, taxonomy

## Abstract

Species of the basidiomycetous genus *Tomentella* are widely distributed throughout temperate forests. Numerous studies on the taxonomy and phylogeny of *Tomentella* have been conducted from the temperate zone in the Northern hemisphere, but few have been from subtropical forests. In this study, four new species, *T. casiae*, *T. guiyangensis*, *T. olivaceomarginata* and *T. rotundata* from the subtropical mixed forests of Southwestern China, are described and illustrated based on morphological characteristics and phylogenetic analyses of the internal transcribed spacer regions (ITS) and the large subunit of the nuclear ribosomal RNA gene (LSU). Molecular analyses using Maximum Likelihood and Bayesian analysis confirmed the phylogenetic positions of these four new species. Anatomical comparisons among the closely related species in phylogenetic and morphological features are discussed. Four new species could be distinguished by the characteristics of basidiocarps, the color of the hymenophoral surface, the size of the basidia, the shape of the basidiospores and some other features.

## 1. Introduction

The genus *Tomentella* Pers. ex Pat. belongs to the family Thelephoraceae Chevall. (Thelephorales, Basidiomycota). Due to *Pseudotomentella* Svrček being merged in *Polyozellus* Murrill recently [1], the Thelephoraceae family comprises the genera *Amaurodon* J. Schröt., *Odontia* Pers., *Polyozellus*, *Tomentella* and *Tomentellopsis* Hjortstam [2,3]. This family mainly presents thin, effused, flabelliform, pileate or resupinate basidiocarps [4]. *Amaurodon* has bluish basidiocarps when fresh that become yellow-green when dry. *Odontia* has a granulose or hydnoid hymenophoral surface and verruculose basidiospores. *Tomentellopsis* is characterized by a dimitic hyphal system with simple-septate generative hyphae and echinulate basidiospores less than 6 µm in length and with an ellipsoidal frontal face [3]. *Polyozellus* has resupinate to erected basidiocarps with matt-appearance hymenia [1,5,6,7]. *Tomentella* produces resupinate basidiocarps that form cottony or spider web-like layers on fallen wood, leaf litter, soil and other substrates [1,2,6,8].

Ectomycorrhizal fungi (EMF) play a significant role in recycling nutrients, digesting plant and insect remnants and maintaining biodiversity in natural ecosystems. EMF produce obviously different types of basidiocarps, which are conspicuous or inconspicuous basidiocarps. The genus *Tomentella* is a widely distributed ectomycorrhizal lineage [9,10] and can form ectomycorrhiza with different host tree families, e.g., Asteropeiaceae, Dipterocarpaceae, Fabaceae, Gnetaceae, Pinaceae, Proteaceae, Sapotaceae, Sarcolaenaceae and Phyllanthaceae [2,11,12,13,14,15,16,17,18,19,20]. Since the first finding of an ectomycorrhiza formed by a *Tomentella* species [11], many studies have confirmed the *Tomentella*-*Thelephora* to play an important role in receiving energy and transporting nutrients to their host plants [11,12,13,21,22,23,24,25,26,27]. *Tomentella* play a protective and promoting role for growth and development in extreme environments [28]. In addition, the *Tomentella*-*Thelephora* lineage reading from the mycorrhizal root tips of Betulaceae, Fagaceae, Pinaceae and Tiliaceae account for 38.2% of the total EMF [29]. The *Tomentella*-*Thelephora* lineages are also common in pine forests and nurseries and are intensively studied as they are often used for seedling inoculation in reforestation programs [30]. Although these studies have revealed the presence of a large number of *Tomentella* OTUs in different type of forests, most sequences are difficult to identify at the species level.

There are many studies on species diversity and taxonomy of *Tomentella* reported in temperate forests, but few have been conducted in subtropical and tropical regions [1,2,6,30]. Guizhou Province, known as the “Karst Province”, is located on the Yunnan–Guizhou Plateau, Southwestern China, and has a subtropical humid monsoon climate [31]. The unique terrain, landforms, climate and vegetation create excellent conditions for the growth of macrofungi [32,33]. In 2023, dozens of *Tomentella* specimens were collected from two subtropical forests in Guiyang, Guizhou Province. Changpoling National Forest Park is mainly dominated by coniferous trees such as *Pinus* spp., and a number of broad-leaved trees are scattered in the forests [34]. Qianlingshan Park is the area characterized by karst landform, and dominated by a subtropical evergreen secondary broad-leaved forest, with the domination of *Broussonetia popyifera*, *Cunninghamia lanceolate*, *Pinus massoniana*, *Platycladus orientalis*, *Polygonum barbatum* and *Quercus acutissima* [35].

In this study, four new species were described using morphological and phylogenetic analyses of DNA sequences. The main aim of this study is to update the species diversity of *Tomentella* in subtropical forests of China.

## 2. Materials and Methods

### 2.1. Specimen Collections

Specimens were collected from Changpoling National Forest Park (106°39′10″ E–106°40′10″ E, 26°38′45″ N–26°40′00″ N, Altitude: 1202–1370 m) and Qianlingshan Park (106°41′32″ E, 26°35′53″ N, Altitude: 1100–1396 m), Guizhou Province, Southwestern China. The regions belong to the subtropical humid monsoon climate with the obvious precipitations in summer. The average annual temperature is 13.6 °C, with average annual precipitation of 1200 mm. The specimen information, host trees, ecological habits, collector and date were recorded, and the photos of the basidiocarps and growth environment were taken. Then, the specimens were dried and bagged in time for preservation. The specimens were deposited at the herbarium of the Institute of Applied Ecology, Chinese Academy of Sciences (IFP).

### 2.2. Morphological Studies

Macromorphological characteristics, including the color, texture and thickness of basidiocarps, hymenophoral surface and sterile margin, were examined under a stereomicroscope (Nikon SMZ 1000: Tokyo, Japan) at 4× magnification. The color terms follow Kornerup and Wanscher [36] for the macromorphological description. The microscopic procedure follows Lu [37] with some minor amendments. Microscopic measurements were made from thin sections of basidiocarps mounted in Cotton Blue (abbreviated as CB): 0.1 mg aniline blue dissolved in 60 g pure lactic acid; cyanophilous or acyanophilous reactions were assessed using CB. Amyloid and dextrinoid reactions were tested in Melzer’s reagent (IKI): 1.5 g KI (potassium iodide), 0.5 g I (crystalline iodine), 22 g chloral hydrate, aq. dest. 20 mL, inamyloid = neither amyloid nor dextrinoid reaction. Micromorphological descriptions were studied at magnifications up to 1000× with a light microscope (Nikon Eclipse E600: Tokyo, Japan) with phase contrast illumination, and dimensions were estimated subjectively with an accuracy of 0.2 mm. The following abbreviations are used: L = mean spore length (arithmetical average of all spores), W = mean spore width (arithmetical average of all spores), Q = extreme values of the length/width ratios among the studied specimens and n = the number of spores measured from a given number of specimens. Drawings were made with the aid of a drawing tube. The surface morphology for the basidiospores was observed with a QUANTA 250 scanning electron microscope (ESEM, QUANTA 250, FEI, Eindhoven, The Netherlands) at an accelerating voltage of 25 kV. The working distance was 12.2 mm. A thin layer of gold was coated onto the samples to avoid charging.

### 2.3. DNA Extraction, Amplification and Sequencing

Total genomic DNA was extracted from the dried specimens with a Thermo Scientific Phire Plant Direct PCR Kit (Thermo Fisher Scientific, Waltham, MA, USA). PCR reactions were performed in 30 µL of reaction mixtures containing 15 µL of 2× Phire^®^ Plant PCR buffer, 0.6 µL of Phire^®^ Hot Start II DNA Polymerase, 1.5 µL of each PCR primer (10 mM), 10.5 µL of doubly deionized H_2_O (ddH_2_O) and 0.9 µL of template DNA. For initial species confirmation, the internal transcribed spacer (ITS) region was sequenced for all specimens. The BLAST tool (https://blast.ncbi.nlm.nih.gov/Blast.cgi, accessed on 29 April 2024) was used to compare the resulting sequences with those in GenBank (Table 1). After confirmation of *Tomentella* species, additional gene region coding for the large subunit of nuclear ribosomal RNA gene (LSU) was sequenced. The ITS region was amplified with the primers ITS5 (5′-GGAAGTAAAAGTCGTAACAAGG-3′) and ITS4 (5′-TCCTCCGCTTATTGATATGC-3′) [38,39]. The LSU gene was amplified with the primers LROR (5′-ACCCGCTGAACTTAAGC-3′) and LR7 (5′-TACTACCACCAAGATCT-3′) [40,41]. The PCR thermal cycling program conditions were as follows: initial denaturation at 95 °C for 5 min, followed by 39 cycles at 95 °C for 30 s, × °C (the annealing temperatures for ITS4/ITS5 and LROR/LR7 were 54 °C and 48 °C, respectively) for 30 s [40,42], at 72 °C for 20 s and a final extension at 72 °C for 10 min. All amplified PCR products were estimated visually with 1.4% agarose gels stained with ethidium bromide and sequenced at the Beijing Genomics Institute (BGI) with the same primers.

### 2.4. Phylogenetic Analyses

The new sequences generated in this study were deposited in GenBank. These new sequences, together with the reference sequences of all samples used in the present study, are listed in Table 1. The sequences were edited and condensed with SeqMan v.7.1.0. The sequences generated in this study were supplemented with additional sequences obtained from GenBank (http://www.ncbi.nlm.nih.gov/genbank, accessed on 29 April 2024) and UNITE (https://unite.ut.ee/index.php, accessed on 29 April 2024) based on blast searches and recent publications of the genus *Tomentella*. The sequences were aligned with MAFFT v.7, after which the alignments were manually corrected using MEGA v. 7.0 [43,44]. Phylogenetic analyses including Maximum Likelihood (ML) and Bayesian inference (BI) methods were conducted for the single gene sequence datasets of the ITS and LSU, and the combined dataset of two gene regions. ML analyses were conducted using RAxML-HPC BlackBox 8.2.10 on the CIPRES Science Gateway portal (https://www.phylo.org/portal2, accessed on 29 April 2024) [45], employing a GTRGAMMA substitution model with 1000 bootstrap replicates [46]. BI analyses were conducted using a Markov Chain Monte Carlo (MCMC) algorithm in MrBayes v.3.0 [47]. Two Markov chains were run from a random starting tree for 1,000,000 generations, resulting in a total of 10,000 trees. The first 25% of trees sampled were discarded as burn-in, and the remaining trees were used to calculate the posterior probabilities. Branches with significant Bayesian Posterior Probabilities (BPP > 0.9) were estimated in the remaining 7500 trees. Phylogenetic trees were viewed with FigTree v. 1.4 and processed by Adobe Illustrator CS5.

The ITS sequences of four new species ran UNITE SH matching analyses in PlutoF (https://plutof.ut.ee/, accessed on 29 April 2024) [48,49,50], and the corresponding UNITE SH (species hypothesis) numbers were assigned.

## 3. Results

### 3.1. Phylogenetic Analyses

The combined two-gene sequences dataset (ITS and LSU) was analyzed to determine the phylogenetic positions of the new samples obtained in this study. A total of 2217 characters, including gaps (811 for ITS and 1406 for LSU), were included in the dataset used in the phylogenetic analyses. Of these characters, 928 were constant, 470 were parsimony-uninformative variable and 819 were parsimony-informative. The multiple sequence alignment included the following: nine sequences of the four new species, 211 sequences of *Tomentella* species [51,52,53,54,55,56,57,58,59,60,61,62,63] and one outgroup sequence of *Odontia ferruginea* from Estonia [38,64].

A similar topology was obtained using ML and Bayesian analyses with one sample of *Odontia ferruginea* as the outgroup [65], and only the ML tree is shown in Figure 1 with the ML bootstrap values and Bayesian posterior probabilities. In the phylogenetic tree, 18 clades with moderate to strong support were marked, of which six clades (clade 2, 4, 6, 8, 10 and 16) were consistent with the previous ITS + LSU phylogenetic analyses [65]. Five of the six clades were more strongly supported (100% ML/1.00 BPP for clade 2, 89% ML/1 BPP for clade 4, 100% ML/1.00 BPP for clade 6, 90% ML/0.98 BPP for clade 8 and 84% ML for clade10, respectively), and clade 16 lacked significant support. The phylogenetic tree shows that the nine specimens formed four single clades with strong support (100% ML/1.00 BPP for *T. casiae*, *T. guiyangensis*, *T. olivaceomarginata* and *T. rotundata*) and clustered in the clade that was composed of most species of *Tomentella* used in this study.

### 3.2. Taxonomy

***Tomentella casiae*** H.S. Yuan & Y.Q. Zhu, sp. nov. (Figure 2A, Figure 3 and Figure 4)

Fungal Names: FN 571968

Diagnosis: Hymenophoral surface grayish to gray; sterile margin grayish. Hyphae system monomitic, generative hyphae clamped. Rhizomorphs present, single hypha clamped and rarely simple-septate. Basidiospores subglobose to lobed, echinuli up to 1 µm long.

Type: CHINA. Guizhou Province, Guiyang City, Qianlingshan Park, on fallen angiosperm branch, 21 August 2023, Yuan 18263 (holotype IFP 019963), UNITE SH: SH0917753.10FU.

Etymology: *Casiae* (combinated by CAS and IAE), commemorating the 70th anniversary of the Institute of Applied Ecology (IAE), Chinese Academy of Sciences (CAS).

Basidiocarps: annual, resupinate, separable from the substrate, mucedinoid, without odor or taste when fresh, 0.5–1 mm thick, continuous. Hymenophoral surface granulose, grayish to gray (7D1–7E1) and concolorous with the subiculum; sterile margin often indeterminate, byssoid, lighter than hymenophore, light gray.

Rhizomorphs: present in subiculum and margins, 20–55 µm diam; rhizomorphic surface more or less smooth; hyphae in rhizomorph monomitic, undifferentiated, of type B (according to Agerer, 1987–2008), compactly arranged and uniform; single hypha clamped and rarely simple-septate, thick-walled, unbranched, 1.5–2 µm diam, grayish brown in KOH, acyanophilous, inamyloid.

Subicular hyphae: monomitic; generative hyphae clamped, thick-walled, 3–6 µm diam, without encrustation, brown in KOH and distilled water, cyanophilous, inamyloid. Subhymenial hyphae clamped, slightly thin-walled, 2.5–5 µm diam, without encrustation; hyphal cells more or less uniform, brown in KOH and in distilled water, cyanophilous, inamyloid.

Cystidia: absent.

Basidia: 30–55 µm long, 4–6.5 µm diam at apex and 3–4.5 µm at base, with a clamp connection at base, utriform, not stalked, sinuous, pale brown in KOH and distilled water, 4-sterigmata; sterigmata 2–4 µm long and 1–1.5 µm diam at base.

Basidiospores: (7.4–)7.8–10.1(–10.5) × (6.3–)6.7–8.8(–9.1) µm, L = 8.85 µm, W = 7.86 µm, Q = 1.01–1.30 (n = 60/2), subglobose to bi-, tri- or quadra-lobed in frontal and lateral face, echinulate, pale brown in KOH and distilled water, acyanophilous, inamyloid; echinuli usually grouped in 2 or more, up to 1 µm long.

Additional specimen (paratype) examined: CHINA. Guizhou Province, Guiyang City, Qianlingshan Park, on fallen angiosperm branch 21 August 2023, Yuan 18254 (IFP 019960).

Notes: In the phylogenetic tree (Figure 1), *Tomentella casiae*, *T. brevisterigmata*, *T. muricata*, *T. storea* and *T. lilacinogrisea* clustered together with moderate support (73% in ML). Among these species, *T. casiae* possesses large size basidiospores, bigger than those of the other species. *T. casiae* also possesses rhizomorphs that are the same as those of *T. brevisterigmata*, but *T. casiae* differs by its mucedinoid basidiocarps [65]. *T. casiae* presents two obvious different characteristics comparing with *T. muricata*: the presence of rhizomorphs in the subiculum and margins and the absence of elongated cystidia [66]. *T. casiae* is differentiated from *T. storea* by its mucedinoid basidiocarps and larger basidia [67]. *T. casiae* can be distinguished from *T. lilacinogrisea* by separable basidiocarps [68].

**Figure 2 jof-10-00440-f002:**
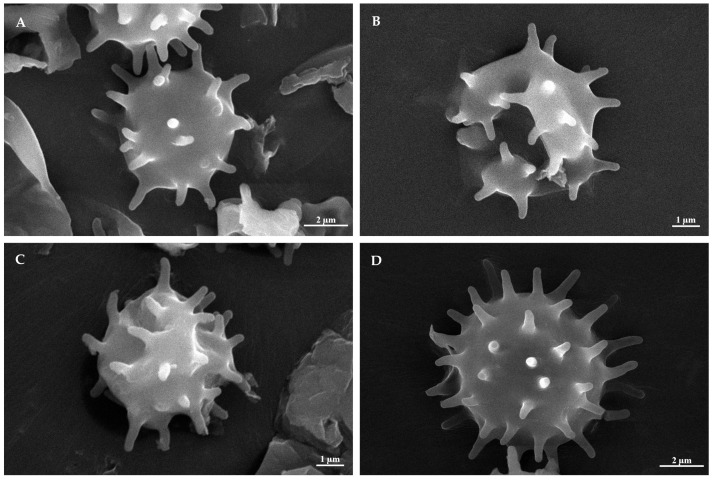
SEM of basidiospores of *Tomentella* species. (**A**) *T. casiae* (holotype Yuan 18263); (**B**) *T. guiyangensis* (holotype Yuan 18281); (**C**) *T. olivaceomarginata* (holotype Yuan 18268); (**D**) *T. rotundata* (holotype Yuan 18269).

**Figure 3 jof-10-00440-f003:**
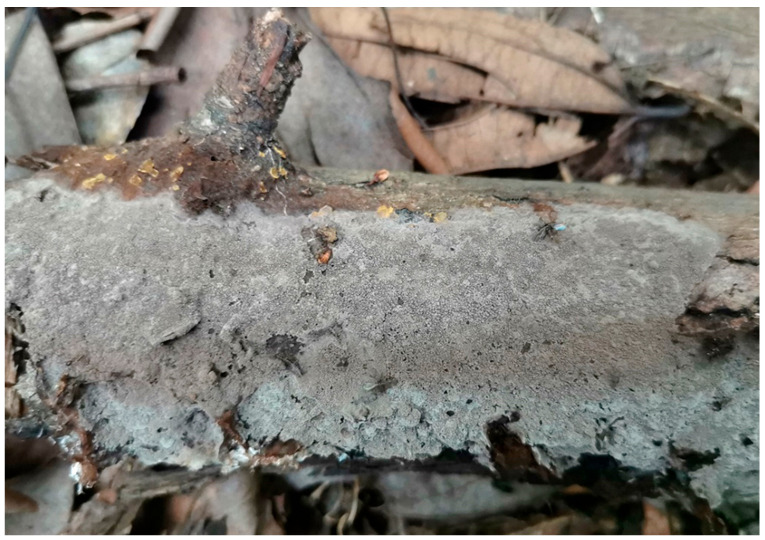
A basidiocarp of *Tomentella casiae* (Holotype Yuan 18263). Photos by Hai-Sheng Yuan.

**Figure 4 jof-10-00440-f004:**
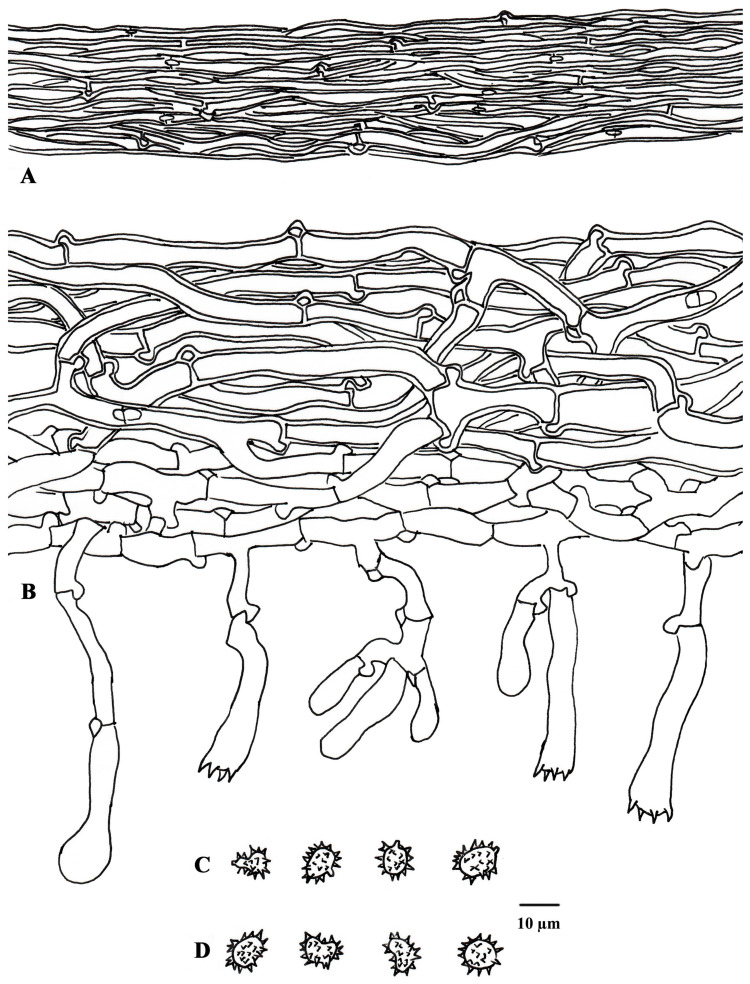
Microscopic structures of *Tomentella casiae* (drawn from holotype Yuan 18263). (**A**) Hyphae from a rhizomorph. (**B**) A section through a basidiocarp. (**C**) Basidiospores in lateral view. (**D**) Basidiospores in frontal view.

***Tomentella guiyangensis*** H.S. Yuan & Y.Q. Zhu, sp. nov. (Figure 2B, Figure 5 and Figure 6)

Fungal Names: FN 571969

Diagnosis: Hymenophoral surface dark brown to chestnut; sterile margin dark brown. Hyphae in rhizomorphs clamped and rarely simple-septate. Basidia short sterigmata. Basidiospores echinuli up to 2 µm long.

Type: CHINA. Guizhou Province, Guiyang City, Qianlingshan Park, on fallen angiosperm trunk, 21 August 2023, Yuan 18281 (holotype IFP 019967), UNITE SH: SH0920617.10FU.

Etymology: *Guiyangensis* (Lat.), named after the collection site of the type specimen, Guiyang City.

Basidiocarps: annual, resupinate, separable from the substrate, mucedinoid, without odor or taste when fresh, 0.6–1.0 mm thick, continuous. Hymenophoral surface smooth, dark brown to chestnut (6E7–6F7) and turning darker than the subiculum when dry; sterile margin often indeterminate, farinaceous, concolorous with hymenophore.

Rhizomorphs: present in subiculum and margins, 10–30 µm diam; rhizomorphic surface more or less smooth; hyphae in rhizomorph: monomitic, undifferentiated, of type B (according to Agerer, 1987–2008), compactly arranged and uniform; single hypha: clamped and rarely simple-septate, thick walled, unbranched, 2–5 µm diam, pale brown in KOH, cyanophilous, inamyloid.

Subicular hyphae: monomitic; generative hyphae: clamped and rarely simple-septate, thick-walled, 3–6 µm diam, with encrustation and grayish brown in KOH and distilled water, acyanophilous, inamyloid. Subhymenial hyphae clamped and rarely simple-septate, thick-walled, 4–6 µm diam, without encrustation; hyphal cells more or less uniform, pale to dark brown in 2.5% KOH and in distilled water, cyanophilous, inamyloid.

Cystidia: absent.

Basidia: 35–55 µm long, 5–9 µm diam at apex and 4–5 µm at base, with a clamp connection at base, clavate, not stalked, sinuous, rarely transverse septa, pale brown in KOH and distilled water, 4-sterigmata; sterigmata 2–3.5 µm long and 1–1.5 µm diam at base.

Basidiospores: (6.3–)6.6–8.5(–8.8) × (4.5–)5.0–7.0(–7.5) µm, L = 7.3 µm, W = 6.1 µm, Q = 1.01–1.67 (n = 60/2), subglobose to bi-, tri- or quadra-lobed in frontal and lateral face, echinulate to aculeate, pale brown in KOH and distilled water, acyanophilous, inamyloid; echinuli usually grouped in 2 or more, up to 2 µm long.

Additional specimens (paratype) examined: CHINA. Guizhou Province, Guiyang City, Qianlingshan Park, on fallen angiosperm trunk, 21 August 2023, Yuan 18256 (IFP 019961); on fallen branch of *Quercus*, 21 August 2023, Yuan18259 (IFP 019962).

Notes: In the phylogenetic tree (Figure 1), *Tomentella guiyangensis* and *T. parmastoana* clustered together with significant support (100% in ML and 1.00 BPP). They exhibit some similar characteristics: mucedinoid basidiocarps, more or less similar color of the hymenophoral surface (brown, grayish brown or dark brown), monomitic rhizomorphs and size of basidiospores [58]. However, *T. guiyangensis* differs from *T. parmastoana* by separable basidiocarps and larger basidia (35–55 µm).

**Figure 5 jof-10-00440-f005:**
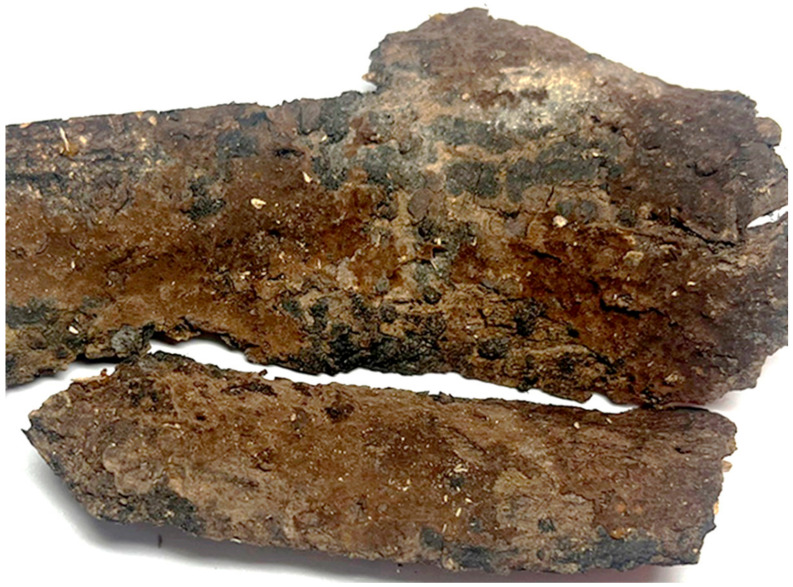
A basidiocarp of *Tomentella guiyangensis* (Holotype Yuan18281). Photos by Ya-Quan Zhu.

***Tomentella olivaceomarginata*** H.S. Yuan & Y.Q. Zhu, sp. nov. (Figure 2C, Figure 7 and Figure 8)

Fungal Names: FN 571971

Diagnosis: Hymenophoral surface pale brown to brown; sterile margin olivaceous. Hyphae in rhizomorphs simple-septate; generative hyphae clamped. Basidiospores subglobose to bi-, tri- or quadra-lobed in frontal and lateral face.

Type: CHINA. Guizhou Province, Guiyang City, Qianlingshan Park, on fallen angiosperm branch, 21 August 2023, Yuan 18268 (holotype IFP 019964), UNITE SH: SH0921288.10FU.

Etymology: *Olivaceomarginata* (Lat.), referring to the brown hymenophoral surface and olive sterile margin.

Basidiocarps: annual, resupinate, separable from the substrate, arachnoid or mucedinoid, without odor or taste when fresh, 1.5–3 mm thick, continuous. Hymenophoral surface smooth, pale brown to brown (5D4–6E7) and turning darker than subiculum when dry; sterile margin often indeterminate, farinaceous, paler than hymenophore, olivaceous.

Rhizomorphs: present in subiculum and margins, 30–45 µm diam; rhizomorphic surface: more or less smooth; hyphae in rhizomorph: monomitic, undifferentiated, of type B (according to Agerer, 1987–2008), compactly arranged and uniform; single hypha: simple-septate, thick walled, unbranched, 3–5 µm diam, brown in KOH, acyanophilous, inamyloid.

Subicular hyphae: monomitic; generative hyphae clamped, slightly thick walled, 4–8 µm diam, with encrustation, pale to dark brown in KOH and distilled water, acyanophilous, inamyloid. Subhymenial hyphae clamped, thin-walled, 4–6 µm diam, without encrustation; hyphal cells more or less uniform, pale to dark brown in 2.5% KOH and in distilled water, acyanophilous, inamyloid.

Cystidia: absent.

Basidia: 15–35 µm long, 6–8 µm diam at apex and 3–5 µm at base, with simple septa at base, clavate, not stalked, without transverse septa, pale brown in KOH and distilled water, 4-sterigmata; sterigmata: 3–4 µm long and 1.5–2 µm diam at base.

**Figure 6 jof-10-00440-f006:**
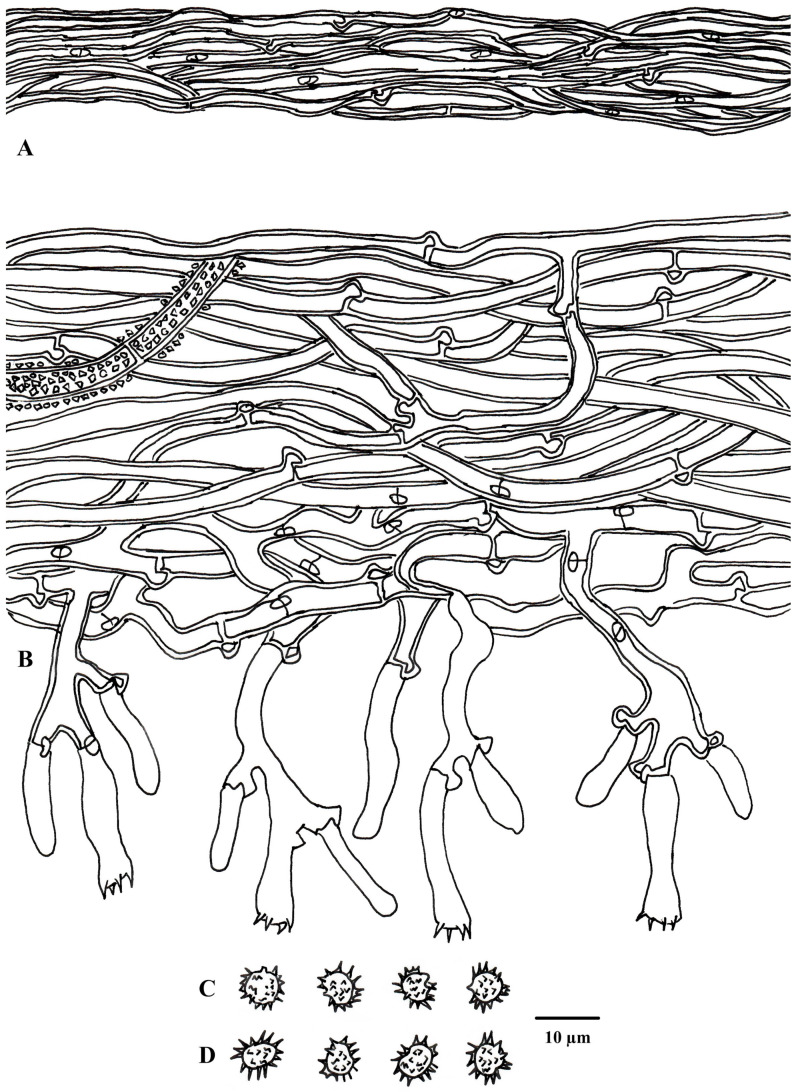
Microscopic structures of *Tomentella guiyangensis* (drawn from holotype Yuan 18281). (**A**) Hyphae from a rhizomorph. (**B**) A section through a basidiocarp. (**C**) Basidiospores in lateral view. (**D**) Basidiospores in frontal view.

Basidiospores: (5.1–)6–7.7 (–8.1) × (4.5–)4.8–7(–7.5) µm, L = 6.75 µm, W = 5.74 µm, Q = 1.01–1.60 (n = 60/2), subglobose to bi-, tri- or quadra-lobed in frontal and lateral face, echinulate to aculeate, grayish brown in KOH and distilled water, cyanophilous, inamyloid; echinuli or aculei usually grouped in 2 or more, up to 1 µm long.

Additional specimen (paratype) examined: CHINA. Guizhou Province, Guiyang City, Qianlingshan Park, on rotten wood of *Pinus massoniana*, 21 August 2023, Dai 25782.

Notes: *Tomentella olivaceomarginata* and *T. farinosa* showed a close relationship with significant support (100% in ML and 0.99 BPP) in the phylogenetic tree (Figure 1), and continuous basidiocarps separable from the substrate, indeterminate sterile margin, clamped hyphae, absence of cystidia, the pale brown to brown hymenophoral surface and the basidiospores of approximately the same shape and size are their common characteristics [67]. However, *T. olivaceomarginata* is differentiated from *T. farinosa* by mucedinoid basidiocarps and the presence of rhizomorphs (3–5 µm in diam) [67]. The subclade of the tree also contains *T. pileocystidiata*, and larger basidiospores and the presence of cystidia can be obviously distinguished from those of *T. olivaceomarginata* [58].

**Figure 7 jof-10-00440-f007:**
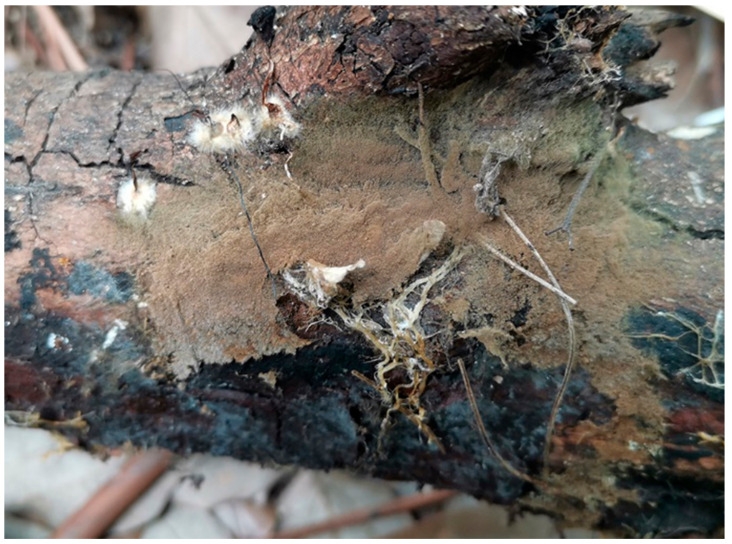
A basidiocarp of *Tomentella olivaceomarginata* (Holotype Yuan 18268). Photos by Hai-Sheng Yuan.

**Figure 8 jof-10-00440-f008:**
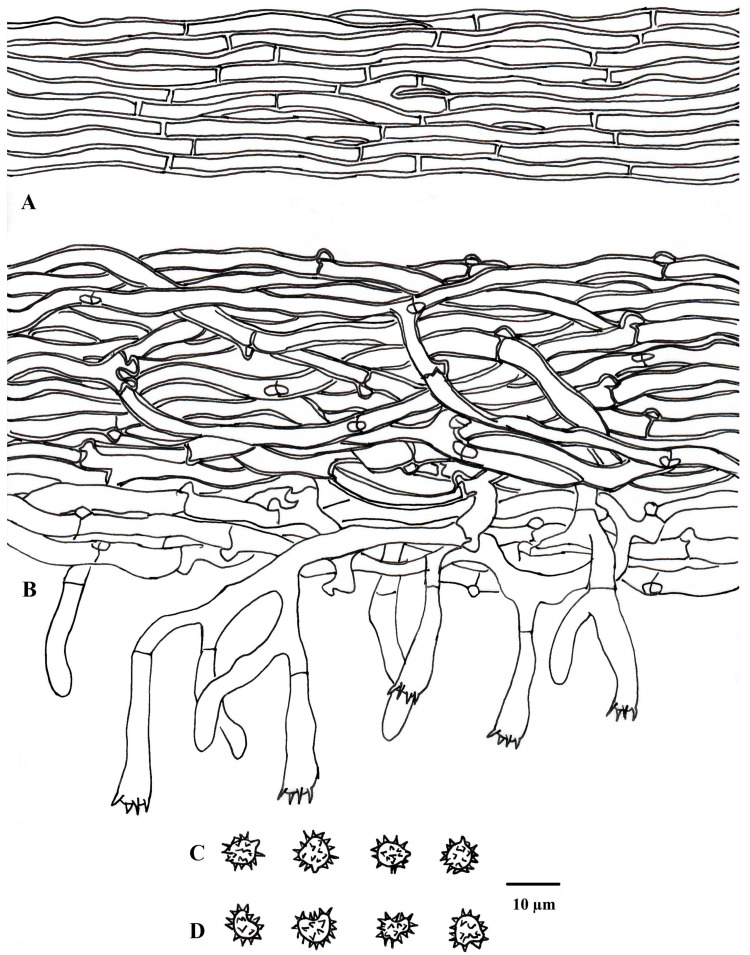
Microscopic structures of *Tomentella olivaceomarginata* (drawn from holotype Yuan 18268). (**A**) Hyphae from a rhizomorph. (**B**) A section through a basidiocarp. (**C**) Basidiospores in lateral view. (**D**) Basidiospores in frontal view.

***Tomentella rotundata*** H.S. Yuan & Y.Q. Zhu, sp. nov. (Figure 2D, Figure 9 and Figure 10)

Fungal Names: FN 571972

Diagnosis: Hymenophoral surface gray to gray brown; sterile margin gray. Rhizomorphs absent; basidia with simple septa at base. Basidiospores echinuli or aculei up to 4 µm long.

Type: CHINA. Guizhou Province, Guiyang City, Qianlingshan Park, on fallen angiosperm branch, 21 August 2023, Yuan 18269 (holotype IFP 019965), UNITE SH: SH0918315.10FU.

Etymology: *Rotundata* (Lat.), referring to the round basidiospores.

Basidiocarps: annual, resupinate, adherent to the substrate, mucedinoid, without odor or taste when fresh, 0.5–0.8 mm thick, continuous. Hymenophoral surface smooth, gray to gray brown (8D3–8E4) and concolorous with the subiculum; sterile margin: often indeterminate, byssoid, concolorous with hymenophore.

Rhizomorphs: absent.

Subicular hyphae: monomitic; generative hyphae clamped and rarely simple-septate, thick-walled, 2–6 µm diam, without encrustation, grayish brown in KOH and distilled water, cyanophilous, inamyloid. Subhymenial hyphae clamped, slightly thick-walled, 4–7 µm diam, without encrustation; hyphal cells short (1–2 µm), brown in KOH and in distilled water, cyanophilous, inamyloid. Cystidia: absent.

Basidia: 15–45 µm long and 5–8 µm diam at apex, 3–4 µm at base, with simple septa at base, utriform, not stalked, sinuous, rarely with transverse septa, pale brown in KOH and distilled water, 4-sterigmata; sterigmata: 9–10 µm long and 2–3 µm diam at base.

Basidiospores: (9.8–)10.2–13.1(–13.5) × (9.2–)9.5–12.1(–12.4) µm, L = 11.8 µm, W = 10.7 µm, Q = 1.00–1.30 (n = 60/2), subglobose to globose in frontal and lateral face, echinulate to aculeate, pale brown in KOH and distilled water, cyanophilous, inamyloid; echinuli or aculei usually isolated, up to 4 µm long.

Additional specimen (paratype) examined: CHINA. Guizhou Province, Guiyang City, Qianlingshan Park, on root of living angiosperm tree, 21 August 2023, Yuan 18273 (IFP 019966).

Notes: *Tomentella rotundata* and *T. pallidocastanea* formed a clade with moderate support (67% in ML) in the phylogenetic tree (Figure 1) and then clustered with *T. guineensis* and *T. maroana* (54% in ML and 0.99 BPP) [69]. Their common features are basidiocarps adherent to the substrate, clamped generative hyphae, utriform basidia and the absence of cystidia. However, *T. rotundata* is distinguished by larger basidiospores (10–13 µm) with longer echinuli (4 μm).

**Figure 9 jof-10-00440-f009:**
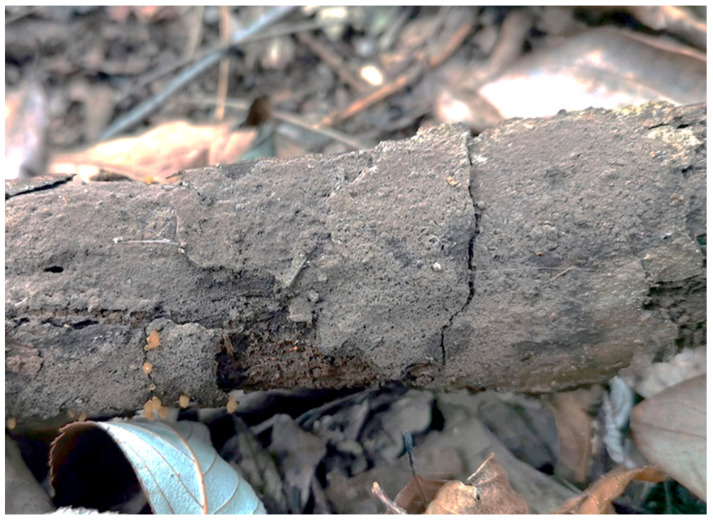
A basidiocarp of *Tomentella rotundata* (Holotype Yuan 18269). Photos by Hai-Sheng Yuan.

**Figure 10 jof-10-00440-f010:**
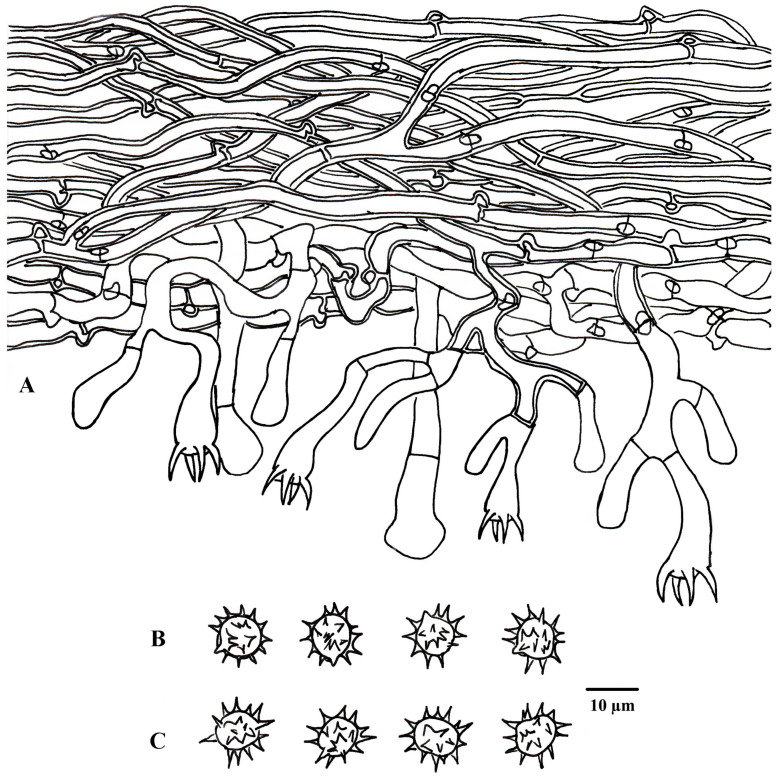
Microscopic structures of *Tomentella rotundata* (drawn from holotype Yuan 18269). (**A**) A section through a basidiocarp. (**B**) Basidiospores in lateral view. (**C**) Basidiospores in frontal view.

## 4. Discussion

In this study, four new *Tomentella* species distributed in Guizhou of Southwestern China were identified by morphological characteristics and phylogenetic analyses combining ITS and LSU sequences. Phylogenetic analyses and morphological features allowed distinguishing the four new species from other known species.

In the phylogenetic tree, 18 clades with moderate to strong support were obtained. Some of these clades were consistent with the previous ITS + LSU phylogenetic analyses [65]. The new species in the clade with high support may share some consistent characteristics. For instance, *Tomentella casiae* fell in clade 7, and the basidiocarps of the species in this clade have an indeterminate sterile margin; *T. rotundata* clustered in clade 16, and species in this clade have clamped and rarely simple-septate subicular hyphae; *T. olivaceomarginata* clustered in clade 17, and species in this clade have separable basidiocarps with an arachnoid surface.

The forests investigated in this study are dominated by coniferous trees *Pinus massoniana* and *P. armandii*, and broad-leaved trees mainly included *Bothrocaryum controversum*, *Celtis sinensis*, *Ligustrum lucidum* and *Robinia pseudoacacia*. The specimens of these four new species were collected from rotten angiosperm wood debris, and their symbiotic plant hosts could not yet be determined. However, the development of resupinate basidiocarps represents an adaptive advantage for species of *Tomentella* growing in primary successional habitats.

In recent years, high-throughput sequencing technology has made it possible to explore unculturable taxa from environmental samples, for example soil or plant root tips, in different regions of the world. However, the BLAST of *Tomentella* species by the ITS sequences in the international gene database (NCBI and UNITE) normally results in a large number of unidentified environmental samples as best matches. It may result from insufficient taxonomic study of this group of fungi. Many studies by high-throughput sequencing have revealed that there are a large number of *Tomentella* spp. in China [62,70,71,72,73,74], and the ectomycorrhizal samples of *Tomentella* have been collected and reported in a previous study from Guizhou Province [75]. Therefore, the species diversity of *Tomentella* in the Karst subtropical forests of Guizhou needs further exploration.

## Figures and Tables

**Figure 1 jof-10-00440-f001:**
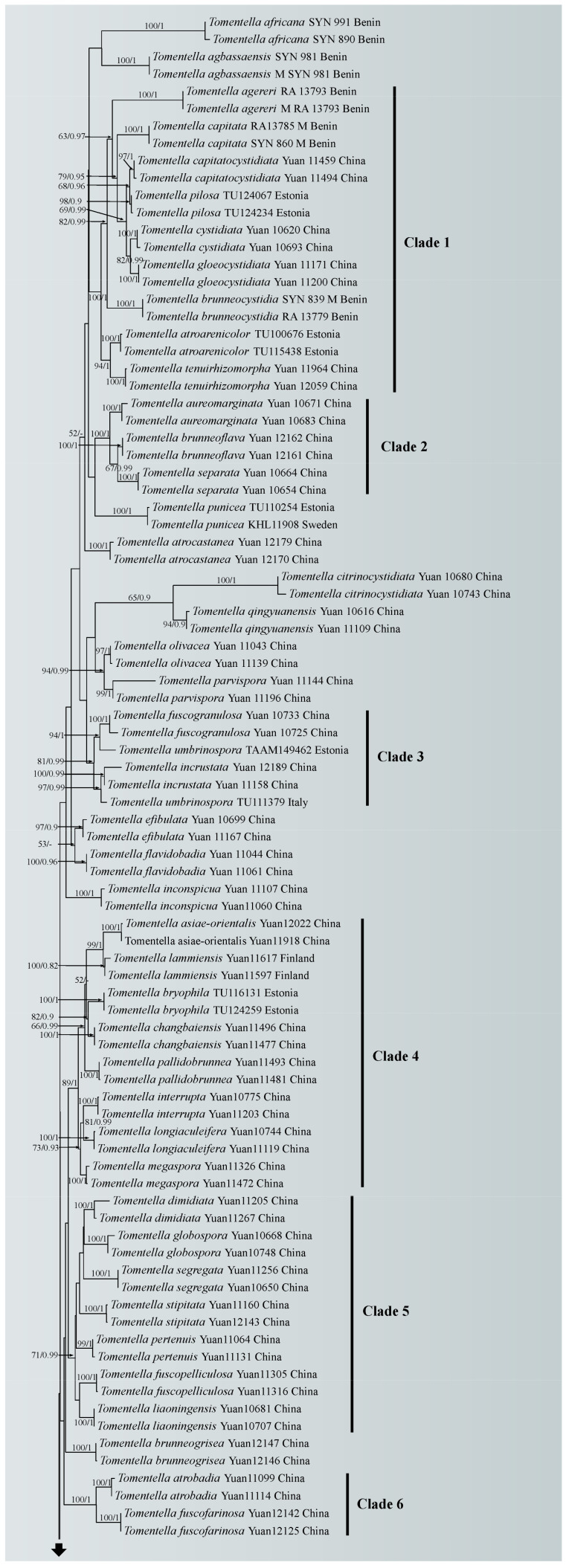
Maximum likelihood tree illustrating the phylogeny of *Tomentella casiae*, *T. guiyangensis*, *T. olivaceomarginata* and *T. rotundata* related taxa based on ITS + LSU nuclear rDNA sequences dataset. Branches are labeled with Maximum likelihood bootstrap equal to or higher than 50% and Bayesian posterior probabilities equal to or higher than 0.9. Vouchers and regions are indicated after the species names. New species in bold (black).

**Table 1 jof-10-00440-t001:** Species, vouchers, GenBank/UNITE accessions and localities of specimens used in this study.

Species	GenBank No./UNITE Database Accession No.		Voucher Number	Substrate	Locality
	ITS	LSU			
*Odontia ferruginea* Pers.	UDB000285	NA	TAAM149492	NA	Estonia
*Tomentella africana* Yorou & Agerer	EF507254	NA	SYN 991	Undersides of dead, burned barks, logs, and leaf litter of native trees	Benin
*T. africana*	EF507256	NA	SYN 890	Undersides of dead, burned barks, logs, and leaf litter of native trees	Benin
*T. afrostuposa* Yorou	JF520431	NA	SYN 2292	On dead valves of *Afzelia africana*	Guinea
*T. afrostuposa*	NR119954	NA	M SYN 2292	On dead valves of *Afzelia africana*	Guinea
*T. agbassaensis* Yorou	NR119638	NA	M SYN 981	On dead valves of *Afzelia africana*	Benin
*T. agbassaensis*	EF507257	NA	SYN 981	Undersides of dead, burned barks, logs, and leaf litter of native trees	Benin
*T. agereri* Yorou	EF538424	NA	RA 13793	On dead valves of *Afzelia africana*	Benin
*T. agereri*	NR119641	NA	M RA 13793	On dead valves of *Afzelia africana*	Benin
*T. alpina* Peintner & Dämmrich	EF655702	NA	IB 20060231	ECM root tips of *Polygonum viviparum*	Austria
*T. alpina*	NR121330	NA	B20060231	ECM root tips of *Polygonum viviparum*	Austria
*T. amyloapiculata* Yorou	EF507263	NA	M SYN 893	On dead valves of *Afzelia africana*	Benin
*T. amyloapiculata*	NA	UDB016726	TU102067	NA	Zambia
*T. asiae-orientalis* H.S. Yuan, X. Lu & Y.C. Dai	MK211711	MK446334	Yuan 12022	On fallen angiosperm twig	China
*T. asiae-orientalis*	MK211710	MK446333	Yuan 11918	On fallen branch of *Pinus koraiensis*	China
*T. asperula* (P. Karst.) Höhn. & Litsch.	NA	UDB018469	TU108147	NA	Estonia
*T. asperula*	KF498576	NA	MT7	*Fagus sylvatica*	Germany
*T. atroarenicolor* Nikol.	NA	UDB018480	TU100676	Under decayed wood	Estonia
*T. atroarenicolor*	NA	UDB016303	TU115438	NA	Estonia
*T. atrobadia* H.S. Yuan & Y.C. Dai	KY686248	MK446335	Yuan 11099	On rotten angiosperm branch	China
*T. atrobadia*	KY686249	MK446336	Yuan 11114	On rotten angiosperm wood debris	China
*T. atrocastanea* H.S. Yuan, X. Lu & Y.C. Dai	MK211743	MK446338	Yuan 12179	On rotten angiosperm wood debris	China
*T. atrocastanea*	MK211742	MK446337	Yuan 12170	On rotten angiosperm wood debris	China
*T. aureomarginata* H.S. Yuan, X. Lu & Y.C. Dai	MK211744	MK446339	Yuan 10671	On rotten angiosperm wood debris	China
*T. aureomarginata*	MK211745	MK878395	Yuan 10683	On rotten angiosperm wood debris	China
*T. badia* (Link) Stalpers	UDB000952	NA	UK427	NA	Estonia
*T. badia*	UDB000238	NA	TAA159022	NA	Russia
*T. beaverae* Suvi & Köljalg	NA	UDB015002	TU105060	*Intsia bijuga*	Seychelles
*T. beaverae*	NA	UDB017787	TU103595	*Intsia bijuga*	Seychelles
*T. bidoupensis* X. Lu & H.S. Yuan	MK775477	MN684329	Yuan 12707	On rotten wood debris of *Pinus kesiya*	Vietnam
*T. bidoupensis*	MK775476	MN684330	Yuan 12685	On rotten wood debris of *Pinus kesiya*	Vietnam
*T. botryoides* (Schwein.) Bourdot & Galzin	UDB000255	AY586717	KHL8453	NA	Sweden
*T. botryoides*	UDB000257	NA	TAAM149614	NA	Russia
*T. bresadolae* (Brinkmann) Bourdot & Galzin	UDB020335	NA	TU115616	NA	Slovenia
*T. bresadolae*	UDB016311	NA	TU115447	NA	Estonia
*T. brevis* H.S. Yuan, X. Lu & Y.C. Dai	MK211746	MK446340	Yuan 11328	On fallen angiosperm branch	China
*T. brevis*	MK211747	MK878396	Yuan 11332	On fallen angiosperm branch	China
*T. brevisterigmata* X. Lu & H.S. Yuan	MK775472	MK850202	Yuan 12700	On rotten wood debris of *Pinus kesiya*	Vietnam
*T. brevisterigmata*	MK775473	MK850203	Yuan 12701	On rotten wood debris of *Pinus kesiya*	Vietnam
*T. brunneocystidia* Yorou & Agerer	DQ848613	NA	SYN 839 (M)	On dead barks and logs of native trees	Benin
*T. brunneocystidia*	DQ848610	NA	RA 13779	On dead barks and logs of native trees	Benin
*T. brunneoflava* H.S. Yuan, X. Lu & Y.C. Dai	MK211749	MK850194	Yuan 12162	On rotten angiosperm wood debris	China
*T. brunneoflava*	MK211748	MK446341	Yuan 12161	On fallen branch of *Larix* sp.	China
*T. brunneogrisea* H.S. Yuan, X. Lu & Y.C. Dai	MK211751	MK446343	Yuan 12147	On fallen angiosperm branch	China
*T. brunneogrisea*	MK211750	MK446342	Yuan 12146	On fallen angiosperm branch	China
*T. brunneorufa* M.J. Larsen	UDB000274	NA	TAAM159857	*Thelephora*-*Tomentella* ECM lineages	Australia
*T. bryophila* (Pers.) M.J. Larsen	UDB014252	NA	TU116131	NA	Estonia
*T. bryophila*	NA	UDB028250	TU124259	NA	Estonia
***T. casiae* H.S. Yuan & Y.Q. Zhu**	**PP479638**	**PP486302**	**Yuan 18263**	**on fallen angiosperm branch**	**China**
** *T. casiae* **	**PP479637**	**PP486299**	**Yuan 18254**	**on fallen angiosperm branch**	**China**
*T. capitata* Yorou & Agerer	DQ848611	NA	RA13785 (M)	On dead bark and log	Benin
*T. capitata*	DQ848612	NA	SYN 860 (M)	On dead bark and log	Benin
*T. capitatocystidiata* H.S. Yuan, X. Lu & Y.C. Dai	MK211700	MK446344	Yuan 11459	On fallen angiosperm branch	China
*T. capitatocystidiata*	MK211701	MK446345	Yuan 11494	On fallen angiosperm branch	China
*T. castanea* (Bourdot & Galzin) Donk	UDB005597	NA	B923	NA	Iran
*T. castanea*	UDB000120	NA	TL-6886	NA	Denmark
*T. changbaiensis* H.S. Yuan, X. Lu & Y.C. Dai	MK211739	MK446347	Yuan 11496	On fallen angiosperm branch	China
*T. changbaiensis*	MK211738	MK446346	Yuan 11477	On fallen angiosperm branch	China
*T. cinerascens* (P. Karst.) Höhn. & Litsch.	NA	UDB016193	TU108037	NA	Estonia
*T. cinerascens*	NA	UDB016498	TU111378	NA	Italy
*T. cinereobrunnea* X. Lu & H.S. Yuan	MK775478	MK850198	Yuan 12703	On rotten wood debris of *Pinus kesiya*	Vietnam
*T. cinereobrunnea*	MK775479	MK850199	Yuan 12705	On rotten wood debris of *Pinus kesiya*	Vietnam
*T. cinereoumbrina* (Bres.) Stalpers	UDB011602	NA	TU115342	NA	Finland
*T. cinereoumbrina*	UDB016491	NA	TU111371	NA	Italy
*T. citrinocystidiata* H.S. Yuan & Y.C. Dai	KY686246	MK446348	Yuan 10680	On rotten angiosperm wood debris	China
*T. citrinocystidiata*	KY686247	MK446349	Yuan 10743	On rotten angiosperm wood debris	China
*T. clavigera* Litsch.	UDB016389	NA	TU115532	NA	Estonia
*T. coerulea* Höhn. & Litsch.	UDB016469	NA	TU115602	NA	Estonia
*T. coerulea*	UDB000266	NA	TAAM153804	NA	Estonia
*T. coffeae* H.S. Yuan & Y.C. Dai	KY686254	MK446350	Yuan 10629	On rotten angiosperm wood debris	China
*T. coffeae*	KY686255	MK446351	Yuan 11100	On fallen angiosperm branch	China
*T. conclusa* H.S. Yuan, X. Lu & Y.C. Dai	MK211703	MK850195	Yuan 12086	On fallen angiosperm branch	China
*T. conclusa*	MK211702	MK446352	Yuan 11986	On fallen trunk of *Pinus koraiensis*	China
*T. cystidiata* H.S. Yuan & Y.C. Dai	KY686219	MK446353	Yuan 10620	On rotten angiosperm wood debris	China
*T. cystidiata*	KY686218	MK446354	Yuan 10693	On rotten angiosperm wood debris	China
*T. dimidiata* H.S. Yuan, X. Lu & Y.C. Dai	MK211704	MK446355	Yuan 11205	On fallen angiosperm branch	China
*T. dimidiata*	MK211705	MK446356	Yuan 11267	On fallen angiosperm branch	China
*T. duplexa* H.S. Yuan, X. Lu & Y.C. Dai	MK211707	MK446358	Yuan 12207	On rotten angiosperm wood debris	China
*T. duplexa*	MK211706	MK446357	Yuan 12202	On rotten angiosperm wood debris	China
*T. efibulata* H.S. Yuan & Y.C. Dai	KY686228	MK446359	Yuan 10699	On rotten angiosperm wood debris	China
*T. efibulata*	KY686229	MK446360	Yuan 11167	On rotten angiosperm wood debris	China
*T. efibulis* H.S. Yuan, X. Lu & Y.C. Dai	MK211708	MK446361	Yuan 11241	On fallen angiosperm branch	China
*T. efibulis*	MK211709	MK446362	Yuan 11329	On fallen angiosperm branch	China
*T. farinosa* H.S. Yuan & Y.C. Dai	KY686251	NA	Yuan 10656	On rotten angiosperm wood debris	China
*T. farinosa*	KY686250	NA	Yuan 10666	On rotten angiosperm trunk	China
*T. flavidobadia* H.S. Yuan & Y.C. Dai	KY686231	MK446364	Yuan 11044	On fallen angiosperm branch	China
*T. flavidobadia*	KY686230	MK446365	Yuan 11061	On fallen angiosperm branch	China
*T. fuscocinerea* (Pers.) Donk	UDB000960	NA	KHL11906	NA	Sweden
*T. fuscocinerea*	UDB000240	UDB018703	TAAM149918	NA	Estonia
*T. fuscocrustosa* H.S. Yuan, X. Lu & Y.C. Dai	MK211713	MK446367	Yuan 11420	On fallen angiosperm branch	China
*T. fuscocrustosa*	MK211712	MK446366	Yuan 11399	On fallen angiosperm branch	China
*T. fuscofarinosa* H.S. Yuan, X. Lu & Y.C. Dai	MK211715	MK446369	Yuan 12142	On fallen angiosperm branch	China
*T. fuscofarinosa*	MK211714	MK446368	Yuan 12125	On fallen angiosperm branch	China
*T. fuscogranulosa* H.S. Yuan & Y.C. Dai	KY686232	MK446370	Yuan 10733	On fallen angiosperm branch	China
*T. fuscogranulosa*	KY686233	MK446371	Yuan 10725	On fallen angiosperm branch	China
*T. fuscopelliculosa* H.S. Yuan, X. Lu & Y.C. Dai	MK211716	MK446372	Yuan 11305	On fallen angiosperm branch	China
*T. fuscopelliculosa*	MK211717	MK446373	Yuan 11316	On fallen angiosperm branch	China
*T. galzinii* Bourdot	UDB000264	NA	RS27093	NA	Finland
*T. galzinii*	HQ204743	NA	2007BBF2	*Quercus ilex*	France
*T. globosa* X. Lu, K. Steffen & H.S. Yuan	MG136838	MH201367	Yuan 11618	On rotten angiosperm wood debris	Finland
*T. globosa*	MG136839	MN684328	Yuan 11603	On rotten angiosperm wood debris	Finland
*T. globospora* H.S. Yuan & Y.C. Dai	KY686242	MK446374	Yuan 10668	On rotten angiosperm wood debris	China
*T. globospora*	KY686243	MK446375	Yuan 10748	On rotten angiosperm wood debris	China
*T. gloeocystidiata* H.S. Yuan & Y.C. Dai	KY686220	MK446376	Yuan 11171	On rotten angiosperm wood debris	China
*T. gloeocystidiata*	KY686221	MK446377	Yuan 11200	On rotten angiosperm wood debris	China
*T. griseocastanea* H.S. Yuan, X. Lu & Y.C. Dai	MK211719	MK446379	Yuan 11409	On fallen angiosperm branch	China
*T. griseocastanea*	MK211718	MK446378	Yuan 11401	On fallen angiosperm branch	China
*T. griseofusca* H.S. Yuan & Y.C. Dai	KY686252	MK446380	Yuan 11094	On fallen angiosperm branch	China
*T. griseofusca*	KY686253	MK446381	Yuan 11104	On rotten angiosperm wood debris	China
*T. griseomarginata* H.S. Yuan, X. Lu & Y.C. Dai	MK211721	MK446383	Yuan 11468	On fallen angiosperm branch	China
*T. griseomarginata*	MK211720	MK446382	Yuan 11458	On fallen angiosperm branch	China
*T. guineensis* Yorou	JF520432	NA	SYN 2331	On dead logs, under *Afzelia africana*	Guinea
*T. guineensis*	NR119955	NA	M SYN 2331	On dead valves of *Afzelia africana*	Guinea
***T. guiyangensis* H.S. Yuan & Y.Q. Zhu**	**PP479645**	**PP486306**	**Yuan 18281**	**on fallen angiosperm trunk**	**China**
** *T. guiyangensis* **	**PP479643**	**PP486300**	**Yuan 18256**	**on fallen angiosperm trunk**	**China**
** *T. guiyangensis* **	**PP479644**	**PP486301**	**Yuan 18259**	**on fallen branch of *Quercus***	**China**
*T. hjortstamiana* Suvi & Köljalg	AM412303	NA	TU103641	*Intsia bijuga*	Seychelles
*T. hjortstamiana*	KC222770	NA	Toohyp24	NA	Australia
*T. inconspicua* H.S. Yuan & Y.C. Dai	KY686234	MK446385	Yuan 11107	On rotten angiosperm branch	China
*T. inconspicua*	KY686235	MK446384	Yuan 11060	On rotten angiosperm wood debris	China
*T. incrustata* H.S. Yuan, X. Lu & Y.C. Dai	MK211723	MK446387	Yuan 12189	On fallen angiosperm branch	China
*T. incrustata*	MK211722	MK446386	Yuan 11158	On fallen angiosperm branch	China
*T. interrupta* H.S. Yu-an & Y.C. Dai	KY686236	MK446388	Yuan 10775	On rotten angiosperm wood debris	China
*T. interrupta*	KY686237	MK446389	Yuan 11203	On rotten angiosperm branch	China
*T. intsiae* Suvi & Köljalg	UDB039732	NA	TU123956	NA	Seychelles
*T. intsiae*	AM412296	NA	TU105130	*Intsia bijuga*	Seychelles
*T. lammiensis* X. Lu, K. Steffen & H.S. Yuan	MG136840	MH201366	Yuan 11617	On rotten angiosperm wood debris	Finland
*T. lammiensis*	MG136841	MH201364	Yuan 11597	On rotten angiosperm wood debris and broad leaf litter	Finland
*T. lapida* (Pers.) Stalpers	NA	UDB016370	TU115604	NA	Estonia
*T. lapida*	NA	UDB016305	TU115440	NA	Estonia
*T. larssoniana* Suvi & Köljalg	NA	UDB017785	TU103690	*Intsia bijuga*	Seychelles
*T. larssoniana*	UDB017790	NA	TU105082	*Intsia bijuga*	Seychelles
*T. lateritia Pat.*	UDB000963	NA	NF S045	NA	Norway
*T. lateritia*	UDB000954	NA	TU108551	NA	Estonia
*T. liaoningensis* H.S. Yuan & Y.C. Dai	MK250814	NA	Yuan 10681	On rotten angiosperm wood debris	China
*T. liaoningensis*	KY686257	NA	Yuan 10707	On fallen angiosperm branch	China
*T. lilacinogrisea* Wakef.	NA	UDB018468	TU108189	NA	Estonia
*T. lilacinogrisea*	NA	UDB016500	TU111381	NA	Italy
*T. longiaculeifera* H.S. Yuan & Y.C. Dai	KY686238	MK446391	Yuan 10744	On bark of fallen angiosperm trunk	China
*T. longiaculeifera*	KY686239	MK446392	Yuan 11119	On rotten angiosperm branch	China
*T. longiechinula* X. Lu & H.S. Yuan	MK775474	MK850201	Yuan 12687	On rotten wood debris of *Pinus kesiya*	Vietnam
*T. longiechinula*	MK775475	MK850200	Yuan 12720	On rotten wood debris of *Pinus kesiya*	Vietnam
*T. longiechinuli* H.S. Yuan, X. Lu & Y.C. Dai	MK211726	MK446393	Yuan 11979	On fallen angiosperm branch	China
*T. longiechinuli*	MK211727	MK446394	Yuan 12083	On fallen angiosperm branch	China
*T. longisterigmata* X. Lu, K. Steffen & H.S. Yuan	MG136836	MN684325	Yuan 11610	On rotten angiosperm wood debris	Finland
*T. longisterigmata*	MG136837	MH201365	Yuan 11602	On rotten angiosperm wood debris	Finland
*T. maroana* Yorou	EF507250	NA	SYN 878	Undersides of dead, burned barks, logs, and leaf litter of native trees	Benin
*T. maroana*	NR119636	NA	M SYN 878	Undersides of dead, burned barks, logs, and leaf litter of native trees	Benin
*T. megaspora* H.S. Yuan, X. Lu & Y.C. Dai	MK211724	MK446395	Yuan 11326	On fallen angiosperm branch	China
*T. megaspora*	MK211725	MK446396	Yuan 11472	On fallen angiosperm branch	China
*T. muricata* (Ellis & Everh.) Wakef.	UDB003303	NA	TU100771	NA	Estonia
*T. muricata*	UDB003310	NA	TU100729	NA	Finland
*T. nitellina* Bourdot & Galzin	EF411085	NA	L2AA1	*Quercus wislizeni*	USA
*T. nitellina*	DQ974778	NA	src675	*Quercus douglasii*	USA
*T. olivacea* H.S. Yuan & Y.C. Dai	KY686224	MK446397	Yuan 11043	On rotten angiosperm branch	China
*T. olivacea*	KY686225	MK446398	Yuan 11139	On rotten angiosperm branch	China
*T. olivaceobrunnea* H.S. Yuan, X. Lu & Y.C. Dai	MK211728	MK446399	Yuan 11194	On rotten angiosperm wood debris	China
*T. olivaceobrunnea*	MK211729	MK446400	Yuan 12148	On rotten angiosperm wood debris	China
***T. olivaceomarginata* H.S. Yuan & Y.Q. Zhu**	**PP479639**	**PP486303**	**Yuan 18268**	**on fallen angiosperm branch**	**China**
** *T. olivaceomarginata* **	**PP479640**	**NA**	**Dai 25782**	**on rotten wood of *Pinus massoniana***	**China**
*T. pallidobrunnea* H.S. Yuan, X. Lu & Y.C. Dai	MK211731	MK446402	Yuan 11493	On rotten angiosperm branch	China
*T. pallidobrunnea*	MK211730	MK446401	Yuan 11481	On fallen angiosperm branch	China
*T. pallidocastanea* X. Lu, Y.H. Mu & H.S. Yuan	MG799183	MN684323	Yuan 11416	On rotten angiosperm wood debris	China
*T. pallidocastanea*	MG816514	MN684324	Yuan 12034	On rotten angiosperm wood debris	China
*T. pallidomarginata* H.S. Yuan, X. Lu & Y.C. Dai	MK211733	MK446404	Yuan 11474	On fallen angiosperm branch	China
*T. pallidomarginata*	MK211732	MK446403	Yuan 11404	On fallen angiosperm branch	China
*T. parmastoana* Suvi & Köljalg	NA	UDB016713	TU105091	*Intsia bijuga*	Seychelles
*T. parmastoana*	NA	UDB017782	TU103691	*Intsia bijuga*	Seychelles
*T. parvispora* H.S. Yuan & Y.C. Dai	KY686226	MK446405	Yuan 11144	On fallen angiosperm branch	China
*T. parvispora*	KY686227	MK446406	Yuan 11196	On fallen angiosperm branch	China
*T. patagonica Kuhar & Rajchenb.*	KT032091	KT032103	BAFC52373	On rotten wood under *Nothofagus dombeyi* Mirb.	Argentina
*T. patagonica*	KT032090	KT032102	BAFC52372	On rotten wood under *Nothofagus dombeyi*	Argentina
*T. pertenuis* H.S. Yuan & Y.C. Dai	KY686240	MK446407	Yuan 11064	On rotten angiosperm branch	China
*T. pertenuis*	KY686241	MK446408	Yuan 11131	On rotten angiosperm stump	China
*T. pileocystidiata* Suvi & Köljalg	UDB015029	NA	TU105068	*Intsia bijuga*	Seychelles
*T. pileocystidiata*	UDB017789	NA	TU105054	*Intsia bijuga*	Seychelles
*T. pilosa* (Burt) Bourdot & Galzin	NA	UDB028059	TU124067	NA	Estonia
*T. pilosa*	NA	UDB028227	TU124234	NA	Estonia
*T. pisoniae* Suvi & Köljalg	NA	UDB002643	TU103671	*Pisonia grandis*	Seychelles
*T. pisoniae*	UDB017778	NA	TU103655	*Pisonia grandis*	Seychelles
*T. pulvinulata* Kuhar & Rajchenb	KT032089	NA	BAFC52371	On rotten wood under *Nothofagus dombeyi*	Argentina
*T. pulvinulata*	KT032088	KT032101	BAFC52370	On rotten wood under *Nothofagus dombeyi*	Argentina
*T. punicea* (Alb. & Schwein.) J. Schröt.	NA	UDB008231	TU110254	NA	Estonia
*T. punicea*	UDB000959	NA	KHL11908	NA	Sweden
*T. pyrolae* (Ellis & Halst.) M.J. Larsen	UDB000262	NA	TAAM005998	NA	Switzerland
*T. qingyuanensis* H.S. Yuan & Y.C. Dai	KY686223	MK446409	Yuan 10616	On rotten angiosperm branch	China
*T. qingyuanensis*	KY686222	MK446410	Yuan 11109	On rotten angiosperm wood debris	China
*T. radiosa* (P. Karst.) Rick	NA	UDB014068	TU110022	NA	Ecuador
*T. radiosa*	UDB000964	NA	NF. S010	NA	Norway
***T. rotundata* H.S. Yuan & Y.Q. Zhu**	**PP479641**	**PP486304**	**Yuan 18269**	**on fallen angiosperm branch**	**China**
** *T. rotundata* **	**PP479642**	**PP486305**	**Yuan 18273**	**on root of living angiosperm tree**	**China**
*T. segregate* H.S. Yuan, X. Lu & Y.C. Dai	MK211735	MK446412	Yuan 11256	On fallen angiosperm branch	China
*T. segregata*	MK211734	MK446411	Yuan 10650	On living tree root	China
*T. separata* H.S. Yuan, X. Lu & Y.C. Dai	MK211737	MK850196	Yuan 10664	On fallen angiosperm trunk	China
*T. separata*	MK211736	MK850197	Yuan 10654	On fallen angiosperm branch	China
*T. stipitata* H.S. Yuan, X. Lu & Y.C. Dai	MK211740	MK446413	Yuan 11160	On fallen angiosperm branch	China
*T. stipitata*	MK211741	MK446414	Yuan 12143	On fallen angiosperm branch	China
*T. stipitobasidia* X. Lu & H.S. Yuan	MK775470	MK850204	Yuan 12713	On wood debris of *Pinus kesiya*	Vietnam
*T. stipitobasidia*	MK775471	MK850205	Yuan 12691	On wood debris of *Pinus kesiya*	Vietnam
*T. storea* H.S. Yuan & Y.C. Dai	KY686244	MK446415	Yuan 10623	On rotten angiosperm wood debris	China
*T. storea*	KY686245	MK446416	Yuan 10749	On rotten angiosperm wood debris	China
*T. stuposa* (Link) Stalpers	NA	MK602778	Th0764	NA	Norway
*T. stuposa*	NA	UDB016174	TU115328	NA	Estonia
*T. subclavigera* Litsch.	NA	UDB031979	TU115594	NA	Finland
*T. subclavigera*	NA	UDB031983	TU115593	NA	Finland
*T. subtestacea* Bourdot & Galzin	NA	UDB016180	TU115374	NA	Ukraine
*T. subtestacea*	NA	UDB016340	TU115482	NA	Estonia
*T. tedersooi* Suvi & Köljalg	UDB017781	NA	TU103673	*Pisonia grandis*	Seychelles
*T. tedersooi*	UDB002644	NA	TU103664	*Pisonia grandis*	Seychelles
*T. tenuirhizomorpha* X. Lu, Y.H. Mu & H.S. Yuan	MG799184	MN684326	Yuan 11964	On rotten angiosperm wood debris	China
*T. tenuirhizomorpha*	MG799185	MN684327	Yuan 12059	On rotten angiosperm wood debris	China
*T. tenuissima* Kuhar & Rajchenb	KT032083	NA	FK15011	On rotten wood under *Nothofagus pumilio*	Argentina
*T. tenuissima*	KT032082	KT032100	BAFC52369	Under a cushion of mosses in a pure *Nothofagus pumilio* forest	Argentina
*T. terrestris* (Berk. & Broome) M.J. Larsen	UDB000222	UDB018708	EL9897	NA	USA
*T. terrestris*	UDB003315	NA	TU100886	NA	France
*T. umbrinospora* M.J. Larsen	NA	UDB016499	TU111379	NA	Italy
*T. umbrinospora*	UDB000233	UDB018709	TAAM149462	NA	Estonia
*T. verruculata* X. Lu & H.S. Yuan	MK775469	MN684332	Yuan 12684	On wood debris of *Pinus kesiya*	Vietnam
*T. verruculata*	MK775468	MN684331	Yuan 12680	On wood debris of *Pinus kesiya*	Vietnam
*T. viridula* (Bourdot & Galzin) Svrček	NA	UDB016192	TU108038	NA	Estonia
*T. viridula*	UDB016392	NA	TU115536	NA	Estonia

Note: NA, not applicable. Strains from present study are in bold.

## Data Availability

The sequences from the present study were submitted to the NCBI website (https://www.ncbi.nlm.nih.gov/, accessed on 29 April 2024), and the accession numbers were listed in Table 1.

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
