# Peer review of "Four New Species of Tomentella (Thelephorales, Basidiomycota) from Subtropical Forests in Southwestern China"

_jof, 2024, doi:10.3390/jof10070440_

Round 1

Reviewer 1 Report

Overall this manuscript on four new Tomentella species is in appropriate quality. Descriptions of the new species are adequate and pictures as well as drawings have sufficient quality. However there is a major issue which must be addressed before the manuscript can be accepted.

The UNITE (https://unite.ut.ee/ ) taxon and species hypothesis paradigm (Kõljalg et al 2013 and 2020) became a standard way how fungi are communicated through the ITS sequences. The GenBank and GBIF are using UNITE SH DOI codes as a species identifier mapped against their taxonomic backbone. I suggest that authors of this manuscript will analyse ITS sequences of the new species and run UNITE SH matching analyses in PlutoF (https://plutof.ut.ee/). Closest SH DOI code can be shown in the description of the new species as well as in the Table 1 by adding a new column after the Species column. Most of the other sequences shown on the ITS tree are part of the UNITE database. But there are much more sequences of the Tomentella species from subtropics which certainly makes sense to analyse with your new sequences. You can simply make an account in PlutoF which is free to use and run analyses by clicking on following menu choices: Laboratories=>Analyses Lab=>Sequence analyses=>New. And then choose the easiest analysis type “massBlaster SH matching (BLAST+2.13.0)”.

UNITE SH paradigm papers cited above:

Kõljalg et al. The Taxon Hypothesis Paradigm—On the Unambiguous Detection and Communication of Taxa. Microorganisms. 2020; 8(12):1910. DOI: 10.3390/microorganisms8121910

Kõljalg et al. 2013. Towards a unified paradigm for sequence-based identification of Fungi. Molecular Ecology, DOI: 10.1111/mec.12481

Line 36

Use the genus name Polyozellus (=Pseudotomentella) or justify why you don't accept the name Polyozellus (see Svantesson et al. 2021).

Lines 129-130

UNITE database (https://unite.ut.ee) must be added here too because some of the sequences are downloaded from this resource.

MycoBank numbers are missing in the descriptions of the new species.

Table 1

Add column “UNITE SH” after the column “Species”. The closest SH code of the new species should be added in this column. See remarks in the Major comments.

It must be error that two vouchers, Yuan 18281 and 18256 have the same ITS sequence PP479645:

T. guiyangensis H.S. Yuan & Y.Q. Zhu 

PP479645 

PP486306 

Yuan 18281

on fallen angiosperm trunk 

China 

T. guiyangensis 

PP479645 

PP486300 

Yuan 18256 

on fallen angiosperm trunk 

China 

References

33 . Zhang, Y.Y.; Long, J.; Li, P.; Chen, X.M.; Jiang, Y.C.; Wang, Y.R. Spatial structure of secondary forest in Changpoling Forest 480 Farm of Guiyang. J. Zhejiang Forest. Sci. Technol. 2018, 38, 37–43.

Add note in the end of the reference that this paper is in Chinese

Author Response

Response to Reviewer 1 Comments

Dear reviewer 1,

Thank you for giving us the opportunity to submit a revised draft of the manuscript for publication in the Journal of Fungi. We appreciate the time and effort that you dedicated to providing feedback on our manuscript and are grateful for the insightful comments on and valuable improvements to our paper.

Best wishes.

Authors.

Overall this manuscript on four new Tomentella species is in appropriate quality. Descriptions of the new species are adequate and pictures as well as drawings have sufficient quality. However there is a major issue which must be addressed before the manuscript can be accepted.

The UNITE (https://unite.ut.ee/ ) taxon and species hypothesis paradigm (Kõljalg et al 2013 and 2020) became a standard way how fungi are communicated through the ITS sequences. The GenBank and GBIF are using UNITE SH DOI codes as a species identifier mapped against their taxonomic backbone. I suggest that authors of this manuscript will analyse ITS sequences of the new species and run UNITE SH matching analyses in PlutoF (https://plutof.ut.ee/). Closest SH DOI code can be shown in the description of the new species as well as in the Table 1 by adding a new column after the Species column. Most of the other sequences shown on the ITS tree are part of the UNITE database. But there are much more sequences of the Tomentella species from subtropics which certainly makes sense to analyse with your new sequences. You can simply make an account in PlutoF which is free to use and run analyses by clicking on following menu choices: Laboratories=>Analyses Lab=>Sequence analyses=>New. And then choose the easiest analysis type “massBlaster SH matching (BLAST+2.13.0)”.

UNITE SH paradigm papers cited above:

Kõljalg et al. The Taxon Hypothesis Paradigm—On the Unambiguous Detection and Communication of Taxa. Microorganisms. 2020; 8(12):1910. DOI: 10.3390/microorganisms8121910

Kõljalg et al. 2013. Towards a unified paradigm for sequence-based identification of Fungi. Molecular Ecology, DOI: 10.1111/mec.12481

Point 1: Line 36 Use the genus name Polyozellus (=Pseudotomentella) or justify why you don't accept the name Polyozellus (see Svantesson et al. 2021).

Response 1: Thank you for the genus name suggested. The precedent genus name has been replaced, becoming ‘Polyozellus’.

Point 2: Lines 129-130 UNITE database (https://unite.ut.ee) must be added here too because some of the sequences are downloaded from this resource.

Response 2: Thank you for underlining this deficiency. We added UNITE database (https://unite.ut.ee) to the manuscript.

Point 3: MycoBank numbers are missing in the descriptions of the new species.

Response 3: Thank you for underlining this deficiency. We added Fungal Names numbers.

Point 4: Table 1 Add column “UNITE SH” after the column “Species”. The closest SH code of the new species should be added in this column. See remarks in the Major comments.

Response 4: As suggested by the reviewer, we added the closest SH code of the new species in the manuscript on ‘Taxonomy’ section.

Point 5: It must be error that two vouchers, Yuan 18281 and 18256 have the same ITS sequence PP479645:

Response 5: Thank you for pointing this out. In the revised manuscript, we have corrected the errors accordingly.

Point 6: References

33 . Zhang, Y.Y.; Long, J.; Li, P.; Chen, X.M.; Jiang, Y.C.; Wang, Y.R. Spatial structure of secondary forest in Changpoling Forest 480 Farm of Guiyang. J. Zhejiang Forest. Sci. Technol. 2018, 38, 37–43.

Add note in the end of the reference that this paper is in Chinese.

Response 6: Thank you for underlining this deficiency. In the revised manuscript, we added ‘(in Chinese)’ in the end of the reference.

Zhang, Y.Y.; Long, J.; Li, P.; Chen, X.M.; Jiang, Y.C.; Wang, Y.R. Spatial structure of secondary forest in Changpoling Forest Farm of Guiyang. J. Zhejiang Forest. Sci. Technol. 2018, 38, 37–43. (in Chinese)

Reviewer 2 Report

Basidiomycetous fungi from the genus Tomentella are characterized by a widespread distribution. They form resupinate basidiocarps on various substrates, but also form ectomycorrhizas with plants from different families. In the current manuscript, four new Tomentella species are described based on morphological characteristics and phylogenetic analyses. Each of them has been illustrated in detail. Very valuable SEM photos of basidiospores were presented. The morphologies of each were compared with the most related Tomentella species on the phylogenetic tree. The manuscript presents very valuable scientific results in the taxonomy of Tomentella and should be published in JoF. The manuscript is prepared very carefully, but requires correction of a few typographical errors or a few unclear passages, which may be the result of the English language used. It is advisable to read the text carefully again and take into account the Remarks marked below.

Remarks

Line 28    ex instead of ex.

Line 47 consider revising this sentence - it should be that instead of the ??

Line 47  rather Tomentella-Thelephora (also elsewhere in the text)

Line 66  Karst or karst?

Line 66  it should be Pinus massoniana,

Line 76  it should be 1100-1396 m

Line 155-156  text requires clarification (in the Tomentella species ?)

Line181 MycoBank no. - please complete before publication (for all four new species)

Line 181 and 188 Tomentella casiae. I have doubts whether we can call ‘casiae’ a Latin word. Your explanation suggests that this is a mix of two acronyms (CAS + IAE  = CASIAE), which brings to mind Latin due to the -ae ending. And, coincidentally, casiae is also a real Latin word – a Genetive Singular form of casia – ‘shrub’, but this fact has no importance here. I would suggest changing that.

Line 183, line 196, line 250, line 299, 'single hyphae';   please note Singular - hypha, Plural - hyphae, here it should be Singular

Line 208   10.1(10.5); 8.8(–9.1) – should be uniform throughout the manuscript (the latter form is rather correct)

 Line 227   T. olivaceomarginata – it should be in italic

Line 287 ‘later face’ - do you mean lateral face

Line 318-326 it would be good to clearly indicate which features apply to which fungal species. This is currently not entirely clear.

Line 337 ‘basidia simple septa at base’ - consider revising this text

Line 341 Rotundata - in italic

Line 341 'rotundatus basidiospores'   it is an error and requires correction

Line 364 ‘are the adherent to the substrate of basidiocarps’ – this requires correction

Line 30   hyphal cells short – please specify (give length)

Line 361  casiae – in italic

Line 445  Mycorrhiza - in italic

Author Response

Response to Reviewer 2 Comments

Dear reviewer 2,

Thank you for giving us the opportunity to submit a revised draft of the manuscript for publication in the Journal of Fungi. We appreciate the time and effort that you dedicated to providing feedback on our manuscript and are grateful for the insightful comments on and valuable improvements to our paper.

Best wishes.

Authors.

Basidiomycetous fungi from the genus Tomentella are characterized by a widespread distribution. They form resupinate basidiocarps on various substrates, but also form ectomycorrhizas with plants from different families. In the current manuscript, four new Tomentella species are described based on morphological characteristics and phylogenetic analyses. Each of them has been illustrated in detail. Very valuable SEM photos of basidiospores were presented. The morphologies of each were compared with the most related Tomentella species on the phylogenetic tree. The manuscript presents very valuable scientific results in the taxonomy of Tomentella and should be published in JoF. The manuscript is prepared very carefully, but requires correction of a few typographical errors or a few unclear passages, which may be the result of the English language used. It is advisable to read the text carefully again and take into account the Remarks marked below.

Point 1: Line 28 ex instead of ex.

Response 1: Thank you for pointing this out. In the revised manuscript, we have corrected the errors accordingly.

Point 2: Line 47 consider revising this sentence - it should be that instead of the ??

Response 3: As suggested by the reviewer, we have changed this sentence in the manuscript.

Point 3: Line 47 rather Tomentella-Thelephora (also elsewhere in the text)

Response 4: Thank you for pointing this out. In the revised manuscript, we have corrected the errors accordingly.

Point 5: Line 66 Karst or karst?

Response 5: ‘karst’. In the revised manuscript, we have corrected the errors accordingly.

Point 6: Line 66 it should be Pinus massoniana,

Response 6: Thank you for pointing this out. In the revised manuscript, we have corrected the errors accordingly.

Point 7: Line 76 it should be 1100-1396 m

Response 7: Thank you for pointing this out. In the revised manuscript, we have corrected the errors accordingly.

Point 8: Line 155-156 text requires clarification (in the Tomentella species ?)

Response 8: Thank you for underlining this deficiency. Vouchers and regions are indicated after the species names.

Point 9: Line181 MycoBank no. - please complete before publication (for all four new species)

Response 9: We added Fungal Names numbers in the revised manuscript.

Point 10: Line 181 and 188 Tomentella casiae. I have doubts whether we can call ‘casiae’ a Latin word. Your explanation suggests that this is a mix of two acronyms (CAS + IAE = CASIAE), which brings to mind Latin due to the -ae ending. And, coincidentally, casiae is also a real Latin word – a Genetive Singular form of casia – ‘shrub’, but this fact has no importance here. I would suggest changing that.

Response10: Thank you for pointing this out. The reviewer is correct, ‘casiae’ is not a Latin word. We have deleted ‘(Lat.)’ in the manuscript on ‘Etymology’ section.

Point 11: Line 183, line 196, line 250, line 299, 'single hyphae'; please note Singular - hypha, Plural - hyphae, here it should be Singular

Response 11: Thank you for underlining this deficiency. We have corrected the errors accordingly.

Point 12: Line 208 10.1(10.5); 8.8(–9.1) – should be uniform throughout the manuscript (the latter form is rather correct)

Response 12: Thank you for underlining this deficiency. We have corrected the errors accordingly.

Point 13: Line 227 T. olivaceomarginata – it should be in italic

Response 13: Thank you for pointing this out. We have corrected the errors accordingly.

Point 14: Line 287 ‘later face’ - do you mean lateral face

Response 14: Lateral face. We have repalced the ‘later face’ in the revised manuscript.

Point 15: Line 318-326 it would be good to clearly indicate which features apply to which fungal species. This is currently not entirely clear.

Response 15: Tomentella olivaceomarginata and T. farinosa have common features: the continuous basidiocarps separable from the substrate, indeterminate sterile margin, clamped hyphae, absence of cystidia, the pale brown to brown hymenophoral surface and the basidiospores of approximately the same shape and size.

Point 16: Line 337 ‘basidia simple septa at base’ - consider revising this text

Response 16: We have changed it to ’basidia with simple septa at base’ in the revised manuscript.

Point 17: Line 341 Rotundata - in italic

Response 17: Thank you for pointing this out. We have corrected the errors accordingly.

Point 18: Line 341 'rotundatus basidiospores' it is an error and requires correction

Response 18: ‘round basidiospores’. We have corrected the errors accordingly.

Point 19: Line 364 ‘are the adherent to the substrate of basidiocarps’ – this requires correction

Response 19: Thank you for pointing this out. ‘are the basidiocarps adherent to the substrate’. We have repalced the ‘are the adherent to the substrate of basidiocarps’ in the revised manuscript.

Point 20: Line 30 hyphal cells short – please specify (give length)

Response 20: Thank you for underlining this deficiency. (1–2 µm). We added it into the revised manuscript.

Point 21: Line 361 casiae – in italic

Response 21: We have corrected the errors accordingly.

Point 22: Line 445 Mycorrhiza - in italic

Response 22: We have corrected the errors accordingly.

Round 2

Reviewer 1 Report

Manuscript is updated.

Manuscript is updated.